# Novel Poly-Arginine Peptide R18D Reduces α-Synuclein Aggregation and Uptake of α-Synuclein Seeds in Cortical Neurons

**DOI:** 10.3390/biomedicines13010122

**Published:** 2025-01-07

**Authors:** Emma C. Robinson, Anastazja M. Gorecki, Samuel R. Pesce, Vaishali Bagda, Ryan S. Anderton, Bruno P. Meloni

**Affiliations:** 1Perron Institute for Neurological and Translational Science, Nedlands 6009, Australia; emma.robinson2@my.nd.edu.au (E.C.R.); 23634587@student.uwa.edu.au (V.B.); 2School of Health Sciences, University of Notre Dame, Fremantle 6106, Australia; anastazja.gorecki@nd.edu.au (A.M.G.); ryan.anderton@nd.edu.au (R.S.A.); 3Curtin Health Innovation Research Institute, Curtin University, Bentley 6102, Australia; 4Centre for Neuromuscular and Neurological Disorders, University of Western Australia, Nedlands 6009, Australia; 22907313@student.uwa.edu.au; 5Department of Neurosurgery, Sir Charles Gairdner Hospital, Nedlands 6009, Australia

**Keywords:** Parkinson’s disease, α-synuclein, cationic arginine-rich peptides, R18D, primary cortical neurons, α-synuclein seeds

## Abstract

Background/Objectives: The role of α-synuclein (α-syn) pathology in Parkinson’s disease (PD) is well established; however, effective therapies remain elusive. Two mechanisms central to PD neurodegeneration are the intracellular aggregation of misfolded α-syn and the uptake of α-syn aggregates into neurons. Cationic arginine-rich peptides (CARPs) are an emerging class of molecule with multiple neuroprotective mechanisms of action, including protein stabilisation. This study characterised both intracellular α-syn aggregation and α-syn uptake in cortical neurons in vitro. Thereafter, this study examined the therapeutic potential of the neuroprotective CARP, R18D (18-mer of D-arginine), to prevent the aforementioned PD pathogenic processes through a cell-free thioflavin-T (ThT) assay and in cortical neurons. Methods: To induce intracellular α-syn aggregation, rat primary cortical neurons were exposed to α-syn seed (0.14 μM) for 2 h to allow uptake of the protein, followed by R18D treatment (0.0625, 0.125, 0.25, 0.5 μM), and a subsequent measurement of α-syn aggregates 48 h later using a homogenous time-resolved fluorescence (HTRF) assay. To assess neuronal uptake, α-syn seeds were covalently labelled with an Alexa-Fluor 488 fluorescent tag, pre-incubated with R18D (0.125, 0.25, 0.5 μM), and then exposed to cortical neurons for 24 h and assessed via confocal microscopy. Results: It was demonstrated that R18D significantly reduced both intracellular α-syn aggregation and α-syn seed uptake in neurons by 37.8% and 77.7%, respectively. Also, R18D reduced the aggregation of α-syn monomers in the cell-free assay. Conclusions: These findings highlight the therapeutic potential of R18D to inhibit key α-syn pathological processes and PD progression.

## 1. Introduction

Parkinson’s disease (PD) is the fastest-growing neurodegenerative disorder, affecting 10 million people worldwide, with no treatments available to address its underlying pathology [1,2]. In most PD cases, except for some familial forms, the progressive death of dopaminergic neurons in the substantia nigra is associated with the intracellular accumulation of aggregated α-synuclein (α-syn) protein, which exhibits a prion-like behaviour [3,4]. α-Syn is a neuronal protein abundantly located in CNS presynaptic terminals where it normally regulates neurotransmitter release [2,4]. In PD, due to a complex interplay of environmental and genetic factors [5,6], α-syn misfolds and aggregates into various forms including β-sheet-rich oligomeric and fibrillar structures, which are a major constituent of Lewy bodies, a hallmark of the disease [4,7]. α-Syn aggregates can interact with lipids and increase the permeability of mitochondrial, lysosomal, and vesicular membranes, disrupting ion homeostasis by allowing calcium influx [8,9,10]. This cascade stimulates cell death pathways including the activation of caspase-3, culminating in neuronal loss particularly in the substantia nigra, and thus manifests as motor symptoms such as resting tremors and bradykinesia [1,8,9,10]. Furthermore, aggregated α-syn behaves like a prion and promotes further aggregation of nearby monomeric α-syn [5,11]. Several in vitro and in vivo models have demonstrated the prion-like propagation of pathological α-syn ’seeds’, which induce aggregation of the protein within cells and spread between cells [3,7,11,12,13,14]. It is likely that both free and exosome-containing α-syn aggregates are released and are taken up by neighbouring neurons via endocytosis [3,15,16,17]. The repeated cycles of intracellular α-syn aggregation release and uptake are a proposed mechanism for neuronal death [3,18] and represent potential therapeutic targets to halt or slow PD progression [19].

Current PD treatments target symptomatic relief, hence the need for therapies that combat the central α-syn pathology and thus disease progression [11]. For example, approaches being considered include reducing α-syn expression [20,21,22] through gene silencing [23] or drugs to stimulate its degradation [24]. However, manipulating the synthesis and degradation of α-syn carries risks, including the potential loss of the protein’s normal functions, which may be counterproductive [25]. Other strategies have aimed to directly inhibit α-syn aggregation and uptake. For example, heat shock proteins act as molecular chaperones to assist in the correct folding of α-syn, but their efficacy appears to diminish once aggregation surpasses a certain threshold [26]. Antibodies targeting protein aggregates have also been developed; however, this method has shown limited success in several studies, likely due to antibodies’ inability to effectively cross the blood–brain barrier and reach the CNS [27,28,29,30]. Moreover, reducing α-syn uptake has been investigated using drugs that block receptor-mediated uptake, such as inhibitors of the lymphocyte activation gene-3 cell-surface protein, with conflicting efficacy across various studies [31,32]. Additional studies are therefore warranted to examine and develop more effective therapeutic strategies for reducing α-syn aggregation and uptake.

Peptide-based therapies have the potential to overcome many of the limitations of current and evolving treatments for PD. Specifically, cationic arginine-rich peptides (CARPs) are a novel and emerging class of molecule, with a combination of biological properties unprecedented for a neuroprotective agent [33,34,35]. For example, CARPs can enter the CNS, traverse cell membranes, antagonise calcium influx, target mitochondria, and stabilise proteins [33,35,36,37]. The poly-arginine peptide, R18D (18-mer of D-arginine; net charge +18), a well characterised neuroprotective CARP [35,36,37], may also inhibit α-syn aggregation and/or block the neuronal uptake of aggregates, while also offering neuroprotective properties against PD [36,38]. In addition, R18D is a proteolytically stable D-enantiomer peptide, offering the possibility of an oral formulation, unlike protein- and antibody-based therapies for the disorder. A recent study from our laboratory demonstrated that R18D can interfere with α-syn uptake in STC-1 enteroendocrine cells [38]. In the current study, we examined the therapeutic potential of R18D to inhibit α-syn monomer aggregation, as well as intracellular α-syn aggregation and seed uptake in primary cortical neurons in vitro.

## 2. Materials and Methods

### 2.1. Cell Culture

#### 2.1.1. Establishment of Rat Primary Cortical Neurons

Animal procedures were approved by the Animal Ethics Committee of the University of Western Australia (AE# 2020/ET00022) and adhered to the Animal Welfare Act 2002 (Western Australia) and the Australian Code for the Care and Use of Animals for Scientific Purposes (8th Ed. 2013). Rat primary cortical cultures were established from embryonic day 18–19 Sprague–Dawley rats (Ozgene ARC Pty Ltd., Perth, WA, Australia), as per published protocols [39,40,41]. Briefly, cortical tissue was isolated and dissociated by gentle trituration in 2 mL of dissociation medium (10 units/mL papain (Sigma-Aldrich, St. Louis, MO, USA), 1.3 mM l-cysteine, and 50 units/mL DNase (Sigma-Aldrich)). The dissociated neurons were washed with 13 mL of cold Dulbecco’s modified eagle medium (DMEM; Life Technologies, Carlsbad, CA, USA) supplemented with 10% horse serum (Life Technologies). After centrifugation, the neurons were gently resuspended in Neurobasal (NB; Life Technologies)/2% B27 supplement (B27; Life Technologies) medium.

#### 2.1.2. Neuronal Cell Plating

Neurons in NB/B27 were seeded into wells pre-coated with poly-D-lysine (Sigma-Aldrich; 50 µg/mL for 60 min prior to cell plating). For 96-well plates, ~52,000 neurons were seeded in 120 μL of medium, per well. For glass coverslips (13 mm Dia) in 24-well plates, ~150,000 neurons were seeded in 700 μL of medium, per well. Plated neurons were maintained in a CO_2_ incubator (5% CO_2_, 95% air balance, 98% humidity, 37 °C). All subsequent incubations used the same incubator, unless stated otherwise. Neuronal cultures received fresh NB/B27 culture media on in vitro days 4 (+50/+200 µL 96-/24-well plate) and 8 (+70/+300 µL 96-/24-well plate). Neuronal cultures were used for experiments 10–12 days after plating.

### 2.2. Study Design

This study had three main aims. The first was to establish a neuronal model of intracellular α-syn aggregation using α-syn seeds, which are misfolded aggregates of the protein that act as a template to promote further aggregation of endogenous monomeric α-syn [42]. Secondly, we examined the ability of R18D to inhibit α-syn aggregation in both a cell-free ThT assay, and intracellularly in neurons. Thirdly, we examined the ability of R18D to inhibit the uptake of α-syn seeds into cortical neurons.

#### 2.2.1. Establishment of Neuronal Model of Intracellular α-Syn Aggregation

Recombinant human α-syn protein seeds (ab21881, Abcam, Cambridge, UK) were distributed into 10 μL aliquots and stored at −80 °C until required. Immediately before use, the α-syn protein aliquots were thawed and sonicated for 2 min to achieve a uniform dispersion of the aggregates. Scanning electron microscopy (Section 2.4.1) was used to qualitatively confirm α-syn aggregation induced by exposure to α-syn seeds in a cell-free assay.

To induce intracellular α-syn aggregation, neurons cultured in 96-well plates were exposed to varying concentrations of α-syn seeds (0.03, 0.07, 0.14, and 0.29 μM) in 50 μL of NB/B27 and incubated for 2 h to allow for α-syn seed uptake into the neurons. Following this, an additional 50 μL of NB/B27 was added to the wells, and plates were then incubated for a further 46 h to allow for α-syn seeds to induce endogenous intracellular α-syn aggregation. At 48 h, the experiment reached its endpoint, and the toxicity of the α-syn seed on neurons was measured using a cell death assay (Section LDH Assay). Thereafter, intracellular α-syn aggregation was measured using a homogenous time-resolved fluorescence (HTRF) assay (Section 2.3.2). An α-syn seed concentration of 0.14 µM was selected for subsequent aggregation studies (Section 2.2.2), as this concentration induced significant α-syn aggregation in neurons (Section 3.2).

#### 2.2.2. Assessment of R18D to Inhibit α-Syn Aggregation

Prior to examining R18D, a dose concentration study was performed to determine the suitable concentration range of the R18D peptide that did not cause significant neuronal toxicity. R18D (H-rrrrrrrrrrrrrrrrrr-NH_2_; r = D-arginine) was synthesised by AmbioPharm (Clearwater, SC, USA) and subjected to high-performance liquid chromatography purification, resulting in a purity of 99%. A 500 µM stock solution of R18D was initially prepared in Baxter water and stored at 4 °C. Neurons growing in a 96-well plate were exposed to different concentrations of R18D (0.016, 0.031 0.061, 0.125, 0.25, 0.5, 1, 2, and 4 μM) in 100 μL of NB/B27 for 48 h, with a media-only control. Toxicity was assessed using a cell death assay and a cell viability assay (Section 2.3.1). Following the results from the toxicity assay (Section 3.5), a peptide concentration range of between 0.05 and 0.5 µM was selected for subsequent α-syn aggregation inhibition studies.

To examine the ability of R18D to inhibit the aggregation of α-syn monomers, a cell-free ThT assay was used (Section 2.3.2).

To examine the ability of R18D to inhibit neuronal intracellular α-syn aggregation, the media from neurons growing in a 96-well plate were removed from the wells and replaced with α-syn seeds (0.14 μM) in 50 μL of NB/B27. The plates were incubated for 2 h to allow for α-syn seed uptake. Then, R18D (0.125, 0.25, and 0.5 μM final concentrations) in 50 μL of NB/B27 was added to the wells, and the plates were incubated for a further 46 h. Control groups consisted of neuronal culture wells treated with 100 µL of NB/B27 medium, with and without α-syn seeds. At 48 h, the experiment endpoint, α-syn intracellular aggregates were quantitatively measured using a homogenous time-resolved fluorescence assay kit (Section 2.3.3).

#### 2.2.3. Assessment of R18D to Inhibit α-Syn Seed Uptake

To examine the ability of R18D to inhibit α-syn seed uptake, neurons growing on coverslips in a 24-well plate and α-syn seeds labelled using the Alexa Fluor^®^ 488 Conjugation Kit (Fast)-Lightning-Link^®^ (Abcam, ab236553) were used. The labelling reaction covalently binds the Alexa Fluor^®^ 488 dye to the lysine residues on proteins. Labelled α-syn seed at a concentration of 0.085 μM in 250 μL of NB/B27 was incubated with different R18D concentrations (0.0625, 0.125, 0.25, and 0.5 µM final concentrations) in 250 μL of NB/B27 for 10 min in the CO_2_ incubator. Media in wells were removed, and 500 μL of the α-syn seed/R18D solution was added to each well. Control groups consisted of neuronal culture wells treated with 500 µL of NB/B27 medium, with and without α-syn seeds. After a 24 h incubation, staining techniques and confocal microscopy (Section 2.4.2) were used to qualitatively and quantitively assess α-syn seed uptake.

### 2.3. Cell Assays

#### 2.3.1. Cell Death and Cell Viability Assays

At the experimental endpoint (Section 2.2.1 and Section 2.2.2), neuronal toxicity was assessed using the LDH and MTS biochemical assays described below. When the HTRF assay (Section 2.3.3) was performed, only the LDH was used to assess for cell toxicity because the MTS assay requires the test to be performed on intact cells which are lysed during the HTRF assay.

##### LDH Assay

To assess for cell death, a lactate dehydrogenase (LDH) release cell death assay (#G1780, Promega, Madison, WI, USA) was conducted, as per the manufacturer’s protocol. The assay measures LDH released from the dead cells using a biochemical reaction that produces a detectable colour change proportional to the number of dead cells [43]. At the experimental endpoint, 20 μL of the well’s supernatant was transferred to a round bottom 96-well plate, and then combined with 20 μL of LDH reagent. After a 10 min incubation period at 37 °C and covered from light, absorbance at 490 nm was measured using a plate spectrophotometer (Asys UVM 340, Biochrom, Cambridge, UK) [43]. The LDH assay was conducted using three wells per group and was repeated across three separate experiments.

##### MTS Assay

To assess for cell viability, the MTS (3-(4,5-Dimethylthiazol-2-yl)-5-(3-carboxymethoxyphenyl)-2-(4-sulfophenyl)-2H-tetrazolium; Promega; #G3582) cell viability assay was used as per the manufacturer’s protocol. The MTS assay measures cell viability by detecting the conversion of the MTS tetrazolium salt into a soluble formazan product by metabolically active cells, with the amount of formazan produced being directly proportional to the number of viable cells [44]. The MTS reagent (8 μL) was added to the culture wells and incubated for 1 h, prior to measuring absorbance at 490 nm using a plate spectrophotometer (Asys UVM 340). The MTS assay was conducted using three wells per group and was repeated across three separate experiments.

#### 2.3.2. Thioflavin T (ThT) Assay

The cell-free aggregation of monomeric α-syn induced by α-syn seeds and inhibited by R18D was assessed using a ThT assay. ThT is a benzathiole dye which selectively binds to β-sheet motifs present in α-syn protein aggregates, causing an increase in fluorescence intensity proportional to aggregate concentration, enabling the quantification of α-syn aggregation [45,46,47]. A stock solution of ThT at 1000 μM (Sigma-Aldrich) was prepared in phosphate-buffered saline (PBS, 14040133; Thermofisher Scientific, Waltham, MA, USA), syringe filtered (0.22 μm), and stored in 5 mL aliquots at 4 °C.

α-Syn seed (0.01 μM) was pre-incubated with varying concentrations of R18D (0.05, 0.1, 0.2, and 0.25 μM final concentrations) in PBS for 15 min at room temperature, before the addition of 5 μL per well of (384-well plate) α-syn monomer (10 μM final concentration; Abcam, ab218818) and ThT (25 μM final concentration) in 5 µL PBS. To account for any background and non-specific fluorescence from the assay’s components, controls consisted of α-syn seeds pre-incubated with the vehicle (PBS) and wells consisting of ThT and α-syn monomer. The plate was sealed with a plastic cover and incubated for 24 h at 37 °C in the Cytation 5 (BioTek, Winooski, VT, USA) plate reader. ThT fluorescence (450 nm excitation/485 nm emission) was measured every 20 min, with 20 s of orbital shaking prior to each measurement.

#### 2.3.3. Homogenous Time-Resolved Fluorescence (HTRF)

At the experimental endpoint (Section 2.2.1 and Section 2.2.2), intracellular α-syn aggregates were measured using a HTRF human α-syn aggregation detection kit (6FASYPEG, Revvity, Waltham MA, USA). The HTRF assay, a relatively new method to detect specific proteins in biological samples, was utilised in this study for several reasons. The HTRF assay is relatively easy to perform, with fewer steps compared with more traditional techniques such as Western analysis, provides high sensitivity with low fluorescence background, and is amenable to high sample throughput. The assay was performed in accordance with the manufacturer’s instructions. The assay uses fluorescence resonance energy transfer (FRET) technology which emits fluorescence when donor and acceptor anti-α-syn antibodies are in close proximity (indicating α-syn aggregates) [48]. Briefly, media in the wells were removed, and neurons were lysed with the addition of 50 μL of lysis buffer (diluted in Baxter water and supplemented with a blocking reagent, as per the kit’s instructions) and agitated (300 rpm) for 30 min at room temperature before collecting the cell lysate. Anti-α-syn aggregate antibodies, one labelled with the fluorophore d2 (acceptor) and the other with terbium-cryptate (donor), were combined in a master mix. For each sample, an equal volume (7.5 μL) of the antibody master mix and cell lysate was added to the wells of a 384-well plate. The plate was sealed and incubated at room temperature for 20 h. To account for any background and non-specific fluorescence from the assay’s components, the control consisted of neuronal lysate treated without α-syn seeds. Fluorescent measurements were performed using a CLARIOstar Plus microplate reader (BMG Labtech, Ortenberg, Germany) equipped with an optical filter device that enabled the simultaneous measurement of both the 620 nm terbium-cryptate donor and 665 nm d2 acceptor emissions. The ratio of the two fluorescence measurements (665 nm/620 nm; acceptor/donor × 10^4^) was calculated, with signal intensity directly proportional to the number of α-syn aggregates present in the sample. The HTRF assay used three wells per treatment group and was repeated three times independently.

### 2.4. Microscopy

#### 2.4.1. Scanning Electron Microscopy

Scanning electron microscopy (SEM) was used to qualitatively confirm the ability of α-syn seeds to induce aggregation of the α-syn monomer protein in a cell-free assay (Section 2.2.1). SEM enables the assessment of the morphological architecture of protein aggregates, allowing for the observation of any changes in their size and shape [49,50]. α-syn protein samples were prepared in PBS, and the α-syn seed (0.01 μM) was incubated either alone or with an α-syn monomer (10 μM; ab218818) for 48 h at 37 °C with orbital shaking at 200 rpm. Samples were deposited over clean 13 mm aluminium stubs, air-dried for 48 h, and coated with a 3 nm platinum sputter (Cressington 208HR, Watford, UK). Samples were imaged by trained technicians at 20,000× and 60,000× magnification using the Neon 40EsB Cross-beam FESEM (ARC LE0775553; Manufacturer: Zeiss, Jena, Germany) at the John de Laeter Centre, Curtin University.

#### 2.4.2. Confocal Microscopy

To qualitatively and quantitively examine the ability of R18D to inhibit α-syn seed uptake into neurons (Section 2.2.3), specific staining protocols, followed by confocal microscopy, were utilised. At the experimental endpoint, media in the wells were removed, and neurons on coverslips were fixed with 4% paraformaldehyde for 30 min at room temperature. Thereafter, coverslips were washed three times with PBS containing 0.1% Tween 20 (PBST; 500 µL per 24-well) for 5 min per wash. Next, the neurons were counterstained with DAPI for nuclei and with the ActinRed™ 555 ReadyProbes™ reagent for cytoskeletal protein actin (ThermoFisher Scientific), with a 20 min incubation period in PBS at room temperature with orbital agitation. Coverslips were mounted onto glass slides using the Fluoromount-G™ mounting medium (ThermoFisher Scientific, R37112).

Cells were imaged using a Nikon Ti-E inverted motorised microscope with Nikon A1Si spectral detector confocal system using a Plan Apo VC 100× NA1.4 oil immersion objective lens (Nikon, Tokyo, Japan) with a pinhole radius of 40.0 μm. Slides were excited with 405 nm (violet), 488 nm (blue), and 568 nm (yellow) lasers. Z-stacks were acquired using the NIS-Elements AR (Nikon) by setting the upper and lower limits based on the cellular processes as visualised by ActinRed cytoskeletal stain, with 10 images taken at 1.4 μm steps at 100× magnification. Three z-stack images from random sections of the coverslip were captured for each treatment group. Imaging was completed across three independent experiments.

Fiji/Image J software (Version 1) was used for the qualitative and semi-quantitative analyses of α-syn neuronal uptake and the inhibition of uptake by R18D. Qualitative observations were made by examining both the proximity and visible intensity of the α-syn seed Alexa Fluor^®^ 488 signal relative to the neuronal actin cytoskeleton protein (ActinRed stain) and nucleus (DAPI), when observing either a Z-stack slice or an orthogonal view across treatment groups. For quantitative measurements, square regions of interest (250 × 250 pixels) were defined around each DAPI-stained nucleus. For each of the three images per treatment group, four regions of interest were measured. A fluorescence intensity of 488 nm within these regions was quantified using mean grey value measurements in Image J. Quantitative measurements were conducted on raw, unprocessed images without any adjustments to brightness, contrast, or other image properties to ensure data integrity and avoid introducing biases.

### 2.5. Statistical Analysis

Data were analysed using GraphPad Prism 10.3.1. The mean and the standard error of the mean were calculated for each treatment group. Normality was tested using Shapiro-Wilk, followed by a one-way analysis of variance (ANOVA) or an unpaired T test to determine statistical significance. Post hoc Fisher’s LSD test was performed to determine significant differences between the treatment groups, with *p* < 0.05 values considered statistically significant. Unless otherwise stated, at least three wells were used in all studies, and the different studies were independently repeated a minimum of three times (n ≥ 3).

## 3. Results

### 3.1. α-Syn Seeds Induce α-Syn Monomer Aggregation in a Cell-Free Assay

Scanning electron microscopy allowed for the morphological visualisation of the α-syn seed by itself and the α-syn seed mixed with an α-syn monomer (Figure 1). In the α-syn seed-only images, aggregation appeared as a central cluster, with minimal surrounding structures. In contrast, in the α-syn seed with α-syn monomer images, a notable increase in the number of aggregates was visible. These qualitative observations suggest that α-syn seeds were capable of inducing α-syn monomer protein aggregation in a cell-free environment, as previously described [51].

### 3.2. R18D Reduces α-Syn Monomer Aggregation in a Cell-Free ThT Assay

At all concentrations examined, R18D significantly reduced cell-free α-syn aggregation (Figure 2 and Table 1). R18D at the 0.05, 0.1, 0.2, and 0.25 μM concentrations increasingly inhibited α-syn aggregation by 28.6% (*p* < 0.001), 51.0% (*p* < 0.001), 62.1% (*p* < 0.001), and 61.2% (*p* < 0.001), respectively.

### 3.3. R18D Is Non-Toxic to Cortical Neurons at Concentrations Below 0.5 μM

Cell death (LDH) and viability (MTS) assays were used to examine R18D toxicity in cortical neurons across a range of concentrations (Figure 3). The LDH release assay indicated no toxicity at R18D concentrations ≤ 0.25 μM (Figure 3A). However, at 0.5 μM, R18D caused a modest but significant (*p* < 0.001) increase in cell death, which increased further at the 1, 2, and 4 μM concentrations. A similar trend emerged with the MTS metabolic viability assay (Figure 3B). R18D at concentrations from 0.0015 to 0.125 μM had little to no impact on metabolic viability, while at 0.25 µM, a slight but significant decrease in metabolic activity (*p* < 0.0373) was observed. Largely concentration-dependent reductions in metabolic activity were observed for R18D at 0.5, 1, 2, and 4 μM.

Based on the toxicity data, and to cover a wide dosage range, R18D at 0.0625, 0.125, 0.25, and 0.5 μM were examined in subsequent neuronal α-syn aggregation studies, despite the potential for toxic effects at the higher concentrations.

### 3.4. α-Syn Seeds Induce Intracellular α-Syn Aggregation in Cortical Neurons

To confirm the cell-free assay findings, α-syn seeds were assessed in cortical neurons to determine their ability to induce intracellular α-syn aggregation. After a 48 h exposure period in cortical neurons, α-syn seeds significantly increased intracellular α-syn aggregate levels in a concentration-dependent manner (*p* < 0.001; Figure 4). Based on these findings, an α-syn seed concentration of 0.14 μM was selected for subsequent studies for the intracellular α-syn aggregation model in cortical neurons.

### 3.5. α-Syn Seeds Enter Neurons

The uptake of α-syn seeds into cortical neurons was confirmed through a 3D qualitative analysis of confocal imaging and a quantitative analysis of the regions of interest (Figure 5). Distinct nuclear and cytoskeletal staining in cortical neurons allowed for the detailed visualisation of neuronal architecture. The fluorescence at 488 nm, indicative of labelled α-syn seeds (α-syn+488), was notably increased in the α-syn seed-treated neurons (Figure 5A). In the α-syn seed-treated neurons, the α-syn+488 signal appeared within neuronal boundaries, as seen in the orthogonal views, confirming the uptake and internalisation of α-syn into neurons. Visible colocalization of the cytoskeletal actin protein (ActinRed stain) with α-syn+488 was considered indicative of the presence of α-syn within neurons. Furthermore, a Z-stack montage (Figure 5B) of the α-syn seed-treated neurons demonstrated a visibly strong 488 signal in slice 4, with noticeably reduced fluorescence intensities in slices 2 and 6, reinforcing the localisation of α-syn seeds within neurons and not on the cell surface. Despite the dense dendritic network making individual neuronal borders and the specific location of α-syn seed challenging to distinguish, the α-syn signal appeared predominantly in perinuclear regions and along cytoplasmic processes (Figure 5D).

### 3.6. α-Syn Seeds Induce Toxicity in Cortical Neurons

Cortical neurons treated with increasing concentrations of α-syn seeds exhibited a dose-dependent modest increase in cell death, based on the LDH release assay (Figure 6). Compared to the untreated control, neuronal cell death was significantly elevated at 0.07 μM (*p* = 0.0412), 0.14 μM (*p* = 0.0081), and 0.29 μM (*p* = 0.0003) α-syn seed concentrations. These results indicate that α-syn seeds at a concentration of ≥0.07 μM induce a modest but significant toxicity in cortical neurons after a 48 h exposure duration. A concentration of 0.14 μM was thus chosen for subsequent aggregations studies, as it generated significant α-syn aggregation (Section 3.4), with minimal toxicity.

### 3.7. R18D Reduces Intracellular α-Syn Aggregation in Cortical Neurons

At all concentrations examined, R18D significantly reduced intracellular α-syn aggregation in cortical neurons compared to the α-syn seed-only control (Figure 7 and Table 1). Interestingly, it was observed that R18D inhibition on intracellular α-syn aggregation did not display a typical concentration dose effect. For example, R18D concentrations of 0.0625, 0.125, 0.25, and 0.5 μM inhibited α-syn aggregation by 33.5% (*p* = 0.003), 37.8% (*p* < 0.0001), 26.9% (*p* = 0.0015), and 17.3% (*p* = 0.022), respectively.

### 3.8. R18D Reduces α-Syn Seed Uptake in Cortical Neurons

Confocal imaging and a quantitative analysis of the regions of interest revealed that R18D treatment effectively reduces α-syn seed (α-syn+488) uptake in neurons (Figure 8 and Table 1). In all concentrations of R18D examined (0.125, 0.25, and 0.5 μM), a noticeable reduction in the green 488 signal was observed in single-plane images compared to neurons treated with the α-syn+488 seed alone (Figure 8A). Likewise, Z-stack orthogonal views further indicate reduced 488 signal intensity within neurons treated with R18D, suggesting a reduction in α-syn seed uptake. The quantitative analysis confirmed that R18D at 0.125, 0.25, and 0.5 µM significantly reduced 488 fluorescence intensity by 77.74%, 76.0%, and 67.0%, respectively (*p* < 0.001). Interestingly, a slightly reduced capacity for R18D to inhibit α-syn seed uptake was observed with increasing peptide concentration, but this effect was not statistically significant.

## 4. Discussion

The role of α-syn in the majority of PD cases is well established; however, effective therapies remain elusive [1,4]. In this study, we demonstrated that the novel poly-arginine peptide R18D has the potential to mitigate key mechanisms driving α-syn pathology [35,36,38] and neurodegeneration in PD and warrants further investigation as a disease-modifying therapeutic.

In this study, α-syn seeds induced aggregation in both the cell-free and cortical neuron models (Section 3.1 and Section 3.4). SEM imaging revealed a visible increase in aggregate formation, while the HTRF analysis of cell lysates showed a dose-dependent increase in neuronal α-syn aggregates with increasing α-syn seed concentrations. These findings align with previous observations in primary cortical neurons [42] and lend further support to the prion-like hypothesis of α-syn seed, which states that the α-syn protein induces aggregation in a self-propagating manner within cells [3,42,52]. Pathological α-syn aggregates have been shown to exhibit several prion-like biochemical properties, such as detergent-insolubility and resistance to protease degradation [52,53]. Additionally, various cellular models have demonstrated the prion-like conversion of normal α-syn protein into abnormal forms, following the kinetic principles of nucleation-dependent protein polymerisation where distinct lag, elongation, and stationary phases can be observed [18,52,54,55]. In our models, it is likely that the prion-like α-syn seed acted upon either the exogenously added monomeric α-syn in the cell-free system or the endogenous α-syn within neurons, triggering a cascade of aggregation. An additional mechanism potentially contributing to α-syn aggregate proliferation involves the interaction between α-syn aggregates and cell surface structures, increasing the local concentration and immediate proximity of aggregates, facilitating further aggregation [56]. It is important to note that in the cell-free experiments before SEM imaging, the samples were agitated during incubation (Section 2.4.1), which may have contributed to the aggregation of monomeric α-syn [57]. Future studies should therefore include a monomer-only control to distinguish between seed-induced and agitation-induced α-syn aggregation. In addition, to provide a better understanding of the overall impact of α-syn seed-induced aggregation of monomeric α-syn and to enable comparisons with other studies, a quantitative measurement of this effect would be useful. Despite these limitations, the strong correlation between our cell-free and neuronal model findings support the role of α-syn seeds as potent inducers of α-syn aggregation. This study establishes an effective model of α-syn aggregation implicated in PD pathology, providing a valuable tool for further investigation of disease mechanisms and potential therapeutic interventions.

Our findings contribute to a growing body of evidence demonstrating neuronal uptake of α-syn aggregates [19,58]. We observed a significant uptake in labelled α-syn seeds within neuronal boundaries by the experiment’s 24 h endpoint (Section 3.5). This observation aligns with the current understanding of α-syn pathology transmission, which involves α-syn release into the extracellular space, followed by its uptake by recipient cells [19,59]. Multiple mechanisms facilitate α-syn uptake, with cell surface protein-mediated endocytosis being the most extensively studied [59,60]. Heparan sulphate proteoglycans (HSPGs) are cell surface molecules which play a critical role in facilitating α-syn entry via electrostatic interactions, concentrating α-syn on the cell surface and promoting its endocytosis [61,62]. This mechanism has been demonstrated in a *Caenorhabditis elegans* model, where knockout of HSPG-related genes blocked the uptake of α-syn fibrils [59,61,62]. Beyond HSPGs, other protein receptors, such as lipoprotein receptor-related proteins (LPRs), have been implicated in α-syn uptake [59]. Studies using iPSC-derived neurons and transgenic mouse models [55,63] have provided compelling evidence for LRPs’ involvement. However, protein-mediated pathways are not the sole route of entry. Direct membrane permeation also facilitates α-syn internalisation [8,56]. Notably, studies on primary cortical neurons have shown that the lipophilic regions and rigid core of α-syn oligomers enable strong interactions with the lipid bilayer, allowing the oligomers to embed within and cross the cell membrane independently [56]. Given the complexity of these uptake mechanisms, it is likely that the α-syn seed entry observed in our study resulted from a combination of these pathways. This multifaceted approach to cellular entry underscores the challenges in developing targeted therapies to prevent α-syn propagation in neurodegenerative diseases.

After confirming α-syn seed aggregation and cellular uptake, we next investigated its neurotoxic effects. The LDH assay results demonstrated a modest but significant neuronal toxicity following the relatively short 48 h exposure to α-syn seeds (Section 3.6). This result corroborates with earlier findings from varying cell types including human embryonic kidney cells (HEK-293) [64] and SH-SY5Y human neuroblastoma cell lines [65] as well as in primary neuronal cultures [42], all of which reported cellular toxicity induced by α-syn aggregates. The neurotoxicity of aggregates is consistent with the emerging understanding of the implication of α-syn pathology on neurodegeneration [4]. Extensive research using both in vitro and in vivo models has elucidated several mechanisms underlying α-syn toxicity, with membrane disruption emerging as a large contributor [56]. Direct membrane permeation has been demonstrated to form pores in the lipid bilayer, increasing basal intracellular calcium and ROS and reducing cell viability [8,9,10,56]. Additionally, α-syn aggregates have been shown to induce mitochondrial dysfunction, increase oxidative stress, and impair synaptic function [66]. These disruptions activate neuroinflammatory pathways and cell death mechanisms, leading to further neuronal damage [67]. Our findings thus reaffirm the role of α-syn in neuronal death, a mechanism closely aligned to dopaminergic neurodegeneration in the midbrain and PD progression [4]. Future studies could assess whether R18D reduces the oxidative stress and mitochondrial dysfunction induced by α-syn seeds, especially after extended exposure durations [68].

R18D has been established as a neuroprotective agent, effectively reducing excitotoxic neuronal death induced by glutamic acid and limiting intracellular calcium influx [69]. Importantly, it was found that R18D was non-toxic below concentrations of 0.5 μM, aligning with a previous study in rat primary cortical neurons that showed no peptide toxicity below 1.5 μM. The slight variation in toxicity threshold is likely due to the difference in endpoint (24 h compared to 48 h), underscoring the importance of considering exposure time in toxicity assessments [69].

This study demonstrated that R18D reduced intracellular α-syn aggregation by 37.8% in cortical neurons (Section 3.7) and aggregation in the cell-free model by 62.1% (Section 3.2). This finding aligns with recent studies and was anticipated given the properties of CARPs to inhibit protein misfolding [36,70,71] and the demonstration that the R18D L-isomer inhibits heat-induced lysozyme protein aggregation [70,71,72]. The efficacy of R18D in reducing α-syn aggregation can be attributed to several mechanisms. Arginine, a known chaotrope, acts as a chemical chaperone, helping to prevent protein misfolding by clustering around protein aggregates to prevent their self-assembly [33,35,36,73]. Moreover, the positive charge of R18D likely interacts electrostatically with the negatively charged moieties of α-syn aggregates, potentially neutralising their charge and preventing further templating of misfolding [36]. Previous work in our laboratory has confirmed R18D’s cell penetrative abilities, and it is therefore likely that specific interactions between R18D and α-syn occur intracellularly [38,74]. Additionally, R18D is a D-enantiomer peptide resistant to proteolytic degradation [75,76] which may contribute to its efficacy over the 48 h timepoint [77]. This resistance to proteolytic degradation makes R18D a promising candidate for therapeutic development, especially as an oral or nasal spray formulation.

Lastly, R18D was very effective at reducing α-syn seed uptake into cortical neurons (Section 3.8). It is likely that the positive charge of R18D competed for cellular uptake pathways such as the HSPG pathway [61,62,78]. A study in primary cortical neurons demonstrated that heparin, a negatively charged molecule structurally related to heparan sulphate moieties in HSPGs, competitively inhibited tau protein aggregate binding to HSPGs [61]. It is possible that R18D due to its electrostatic binding to heparan sulphate moieties, like heparin, competitively blocks α-syn seed binding to cell surface HPSGs [61]. Moreover, the charge neutralisation effect, hindering the anionic nature of α-syn aggregates, could have hindered anionic-dependant uptake mechanisms [59]. Overall, R18D has been demonstrated to reduce α-syn seed uptake which could slow or halt the prion-like propagation of α-syn seeds between cells, a critical aspect of PD progression [79].

While our findings demonstrate the potential of R18D as a novel therapeutic for α-syn pathology, further studies are warranted to address the limitations of our study and extend these findings. Although the addition of α-syn seeds successfully induced aggregation in our model, this approach does not fully replicate the endogenous onset of α-syn aggregation seen in vivo [4]. It should also be mentioned that the in vitro assays were not performed in a blind manner, which is a consideration that could be implemented in future studies to avoid any bias in experimental findings. Future studies should also explore more physiologically relevant models, such as patient-derived induced pluripotent stem cells (iPSCs) differentiated into dopaminergic neurons or cell lines engineered to overexpress α-syn, and animal models of PD [80]. These models could provide valuable insights into R18D’s efficacy in situations that more closely mimic the naturally occurring disease state.

Furthermore, our promising findings regarding R18D’s efficacy to prevent α-syn aggregation and uptake suggest its potential as an early intervention tool. This highlights the critical need for improved early diagnostic tools for PD, as R18D may be most effective in preventing dopaminergic neuronal death before α-syn pathology causes irreversible damage. To this end, to predict the potential clinical efficacy of R18D, it will be important to demonstrate in pre-clinical studies that reduced α-syn aggregation correlates to improvements in neuronal viability, functional outcomes, and clinical biomarkers of the disorder. Additionally, assessing longer timepoints in cell culture and in vivo models would provide valuable insights into the transition from initial α-syn aggregation to fibril formation and allow for chronic condition modelling, providing a more comprehensive understanding of R18D’s long-term effects. In addition, further studies examining the R18D-α-syn aggregate interaction could be useful in modifying R18D and its related peptides to improve efficacy [70,81,82].

## 5. Conclusions

This study has demonstrated that R18D effectively reduces both α-syn aggregation and neuronal α-syn seed uptake, suggesting its potential as a therapeutic agent in mitigating the core of PD pathology. By reducing α-syn aggregation, along with its other neuroprotective functions, R18D may help prevent the formation of toxic oligomers and fibrils and neuronal death in PD. Simultaneously, its ability to inhibit α-syn uptake could slow the spread of pathology throughout the brain, a key factor in PD progression. While these results are encouraging, it is important to note that further research is needed to fully elucidate R18D’s mechanism of action and to assess its efficacy in more clinically relevant models of PD.

## Figures and Tables

**Figure 1 biomedicines-13-00122-f001:**
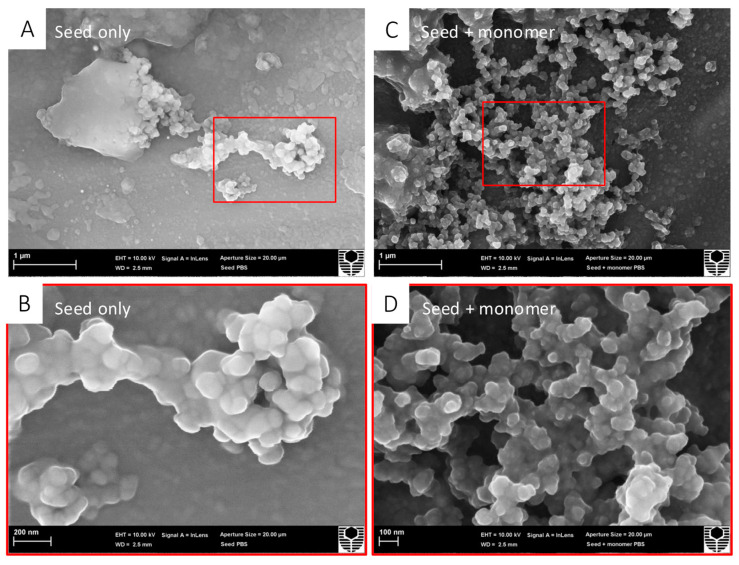
α-Syn seeds induce recombinant human α-syn monomer aggregation in a cell-free assay. Treatment groups were prepared in PBS for SEM; α-syn seeds (0.01 μM) were incubated either alone or with an α-syn monomer (10 μM) at 37 °C degrees for 36 h, with shaking at 200 rpm. Samples were deposited over 13 mm aluminium stubs, air-dried for 48 h, and received a platinum sputter coat prior to SEM. For each treatment group, two magnifications are shown ((**A**,**C**) 20,000×; (**B**,**D**) 60,000×) for each treatment group with the red box indicating the zoomed region. (**A**) α-Syn seed only appearing as a central cluster at 20,000× magnification. (**B**) α-Syn seed only appearing as a central cluster at 60,000× magnification. (**C**) α-Syn seed + α-syn monomer showing an increase in aggregates at 20,000× magnification. (**D**) α-Syn seed + α-syn monomer showing an increase in aggregates at 60,000× magnification.

**Figure 2 biomedicines-13-00122-f002:**
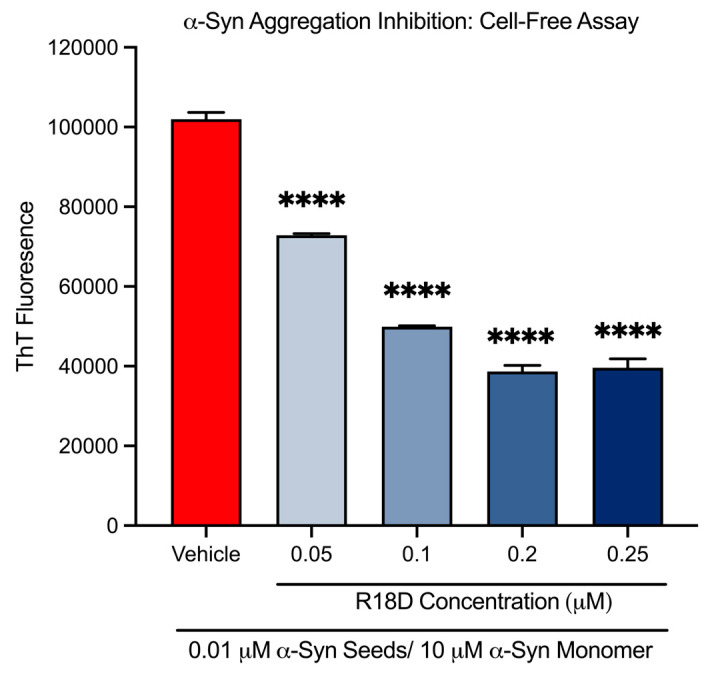
R18D reduces cell-free α-syn aggregation in a cell-free assay. R18D (0.05, 0.1, 0.2, and 0.25 μM) and α-syn seeds (0.01 μM) were pre-incubated for 15 min at room temperature prior to the addition of an α-syn monomer (10 μM), followed by incubation for 24 h at 37 °C. ThT fluorescence values presented are from the 24 h endpoint (450 nm excitation/485 nm emission). Data are mean ± SEM; n = 3. **** signifies *p* < 0.0001 when compared to the vehicle. Vehicle = seed only, no R18D.

**Figure 3 biomedicines-13-00122-f003:**
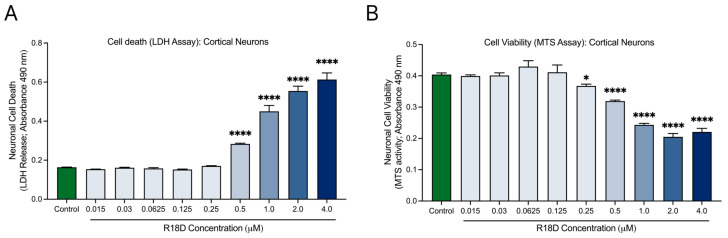
R18D toxicity in cortical neuron cultures. Cortical neurons were incubated with various concentrations of R18D (0.016, 0.031 0.061, 0.125, 0.25, 0.5, 1, 2, and 4 μM) for 48 h. (**A**) Cell death was measured with an LDH release assay. (**B**) Cell viability was measured with an MTS assay. Absorbance values (indicating amount of LDH release or MTS activity) are shown. Data are mean ± SEM; n = 3 from 3 pooled independent experiments. * indicates *p* < 0.05, and **** indicates *p* < 0.0001 when compared to the control. Control = untreated.

**Figure 4 biomedicines-13-00122-f004:**
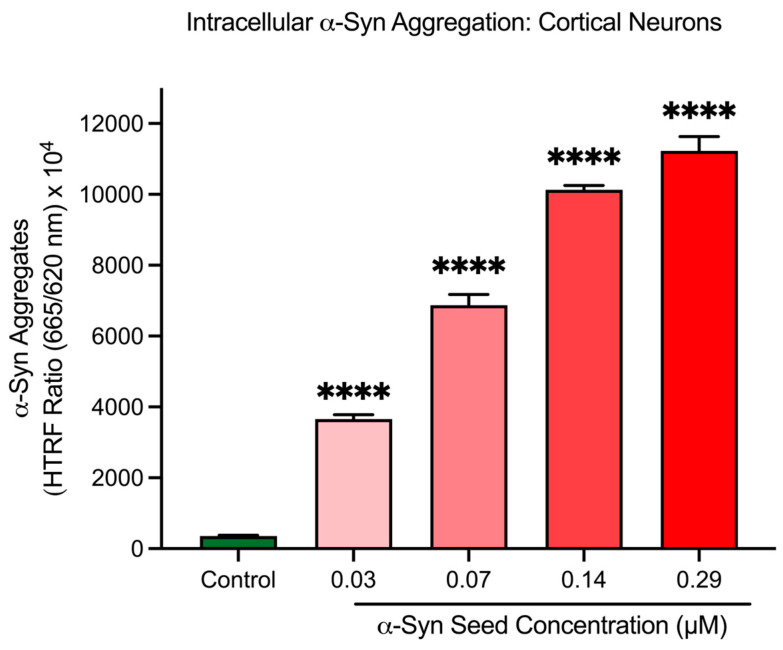
α-Syn seeds induce intracellular α-syn aggregation in cortical neurons. Different concentrations of α-syn seed (0.03, 0.07, 0.14, and 0.29 μM) incubated with cortical neurons for 48 h. Intracellular aggregation measured using an HTRF assay. Fluorescence is presented as a signal ratio at 665 nm/620 nm × 10^4^. Data are shown as mean ± SEM; n = 3 from 3 pooled independent experiments. **** indicates *p* < 0.0001 when compared to the control. Control = untreated.

**Figure 5 biomedicines-13-00122-f005:**
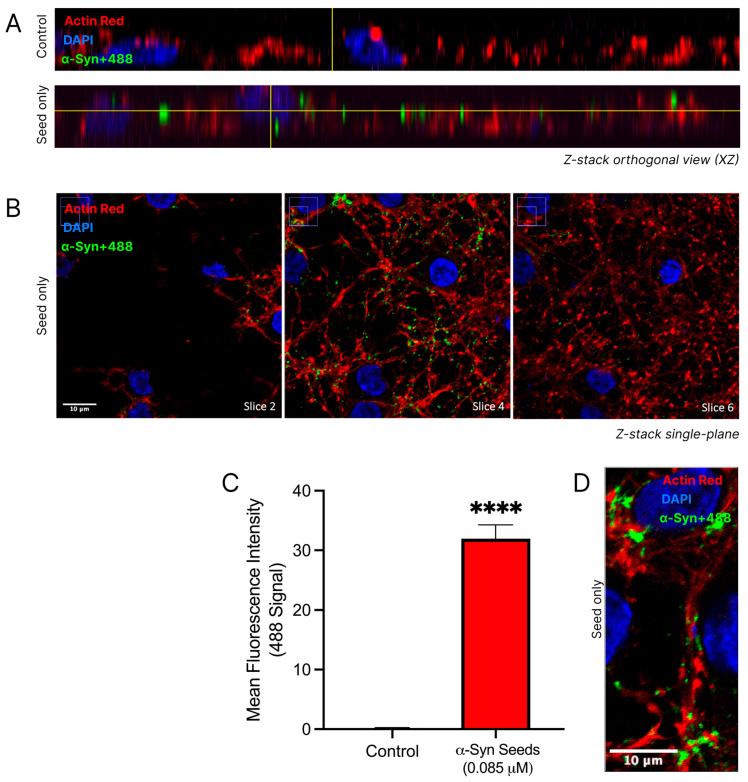
α-Syn seed enters cortical neurons. Cortical neurons were exposed to labelled α-syn seeds (α-syn+488) at 0.085μM, or media only control was exposed to cortical neurons for 24 h. Following treatment, cells were fixed in 4% paraformaldehyde and stained with DAPI (nuclear stain) and ActinRed (cytoskeleton stain) prior to confocal microscopy. α-syn+488 signal indicates the presence of α-syn. (**A**) Orthogonal views of the control and α-syn seed-treated neurons. (**B**) Z-stack montages (slices 2, 4, and 6) at 100× magnification (scale bar = 10 μM) of α-syn seed-treated neurons. Uncropped images provided in Appendix A. (**C**) Quantified 488 tag fluorescence intensity within the square regions of interest (250 × 250 pixels), defined around each DAPI-stained nuclei and attained using the ImageJ/Fiji software. For quantitative analysis, the square regions of interest (250 × 250 pixels) were defined around each DAPI-stained nucleus. For each of the three images per treatment group, four regions of interest were measured. Data are mean ± SEM; n = 4 for each of the 3 images per group, from 3 pooled independent experiments. **** signifies *p* < 0.0001 when compared to the control. Control = untreated. (**D**) A representative single-plane confocal image demonstrating α-syn+488 uptake in cytoplasmic processes (scale bar = 10 μM). Uncropped images provided in Appendix A. Representative confocal images, with minor adjustments made to the brightness and contrast of each channel to improve visualisation (applied across all images identically).

**Figure 6 biomedicines-13-00122-f006:**
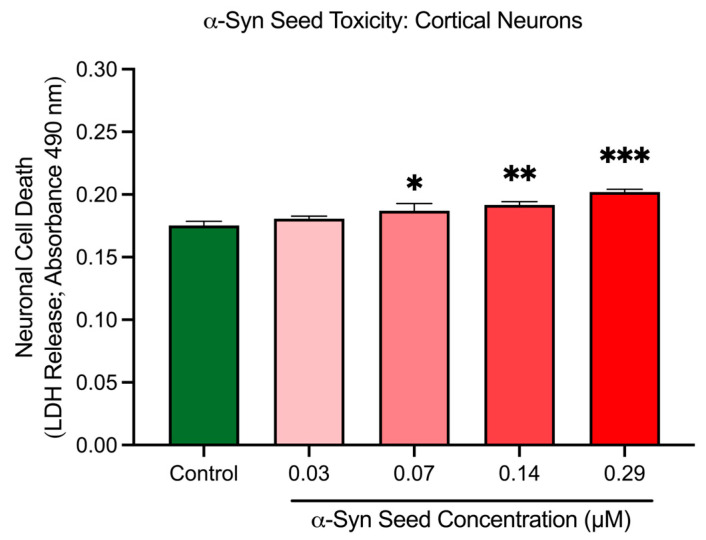
α-Syn seed toxicity in cortical neurons. Cortical neurons were incubated with increasing concentrations of α-syn seeds (0.03, 0.07, 0.14, and 0.29 μM) for 48 h. Cell toxicity (cell death) was measured through an LDH release assay. Data are mean ± SEM; n = 3 from 3 pooled independent experiments. * indicates *p* < 0.05; indicates ** *p* < 0.01, and *** indicates *p* < 0.001 when compared to the control. Control = untreated.

**Figure 7 biomedicines-13-00122-f007:**
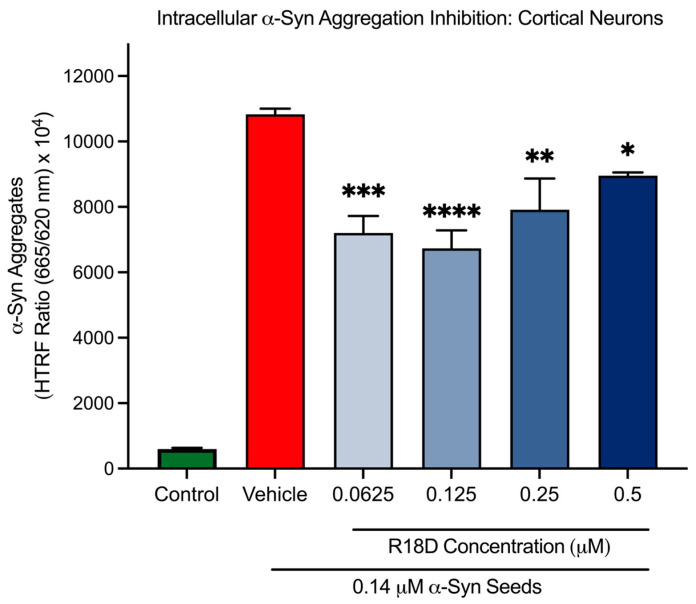
R18D reduces intracellular α-syn aggregation. Cortical neurons were incubated with α-syn seeds (0.14 μM) for 2 h followed by the addition of various concentrations of R18D (0.0625, 0.125, 0.25, and 0.5 µM) with a further incubation for 46 h. Intracellular α-syn aggregation was measured using an HTRF assay. Fluorescence is presented as a signal ratio at 665 nm/620 nm × 10^4^. Data are mean ± SEM; n = 3 from 3 pooled independent experiments. * indicates *p* < 0.05; ** indicates *p* < 0.01; *** indicates *p* < 0.001, and **** indicates *p* < 0.0001 when compared to the vehicle. Vehicle = seed only, no R18D. Control = untreated. Note: Significant difference (*p* < 0.001) was seen between the control and the vehicle.

**Figure 8 biomedicines-13-00122-f008:**
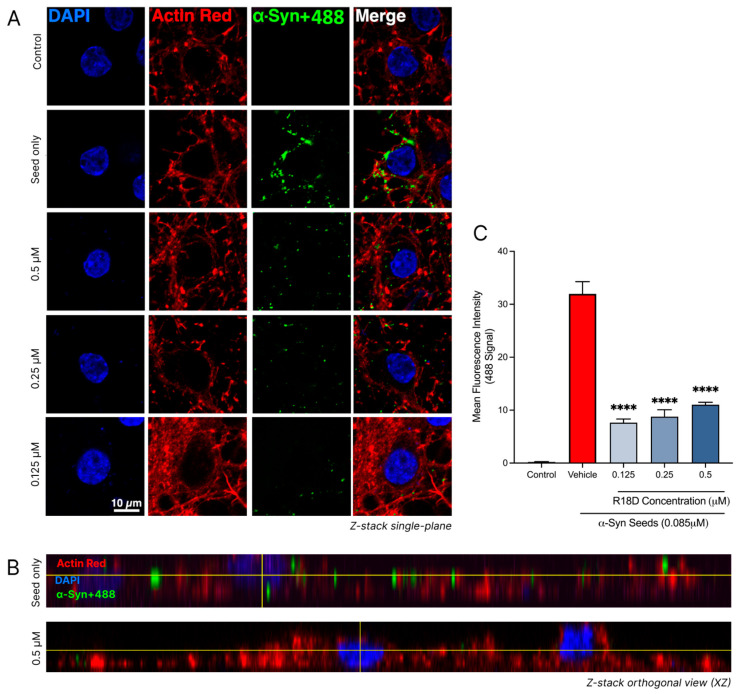
R18D reduces α-syn seed uptake into cortical neurons. α-syn seeds (0.085 μM) were pre-incubated with R18D (0.125, 0.25, 0.5 μM) or media for the α-syn seed-only and control treatments, for 10 min before exposure to cortical neurons for 24 h. Cells were fixed in 4% paraformaldehyde and stained with DAPI (nuclear stain) and ActinRed (cytoskeleton stain) prior to confocal microscopy. (**A**) Representative single-plane confocal images are acquired at 100× magnification (scale bar = 10 μM). Uncropped images provided in Appendix A. (**B**) Orthogonal views (ZY plane) of the neurons exposed to seed with/without R18D. (**C**) Quantified 488 tag green fluorescence intensity within the square regions of interest (250 × 250 pixels), defined around each DAPI-stained nuclei and attained using the ImageJ/Fiji software. For quantitative analysis, the square regions of interest (250 × 250 pixels) were defined around each DAPI-stained nucleus. For each of the three images per treatment group, four regions of interest were measured. Data are mean ± SEM; n = 4 for each of the 3 images per group, from 3 pooled independent experiments. **** indicates *p* < 0.0001 when compared to the vehicle. Vehicle = seed only, no R18D. Control = untreated. Representative confocal images are shown, with minor adjustments made to the brightness and contrast of each channel to improve visualisation (applied across all images identically).

**Table 1 biomedicines-13-00122-t001:** Summary of R18D results.

**Cell-Free ThT α-Syn Mononer Aggregation Assay**		
R18D Concentration Examined	α-Syn Monomer Aggregation—Percentage Inhibition	*p* Value
0.05 μM	28.6%	*p* < 0.001
0.1 μM	51.0%	*p* < 0.001
0.2 μM	62.1%	*p* < 0.001
0.25 μM	61.2%	*p* < 0.001
**Intracellular α-Syn Aggregate HTRFAssay**		
R18D Concentration Examined	α-Syn Aggregation in Neurons—Percentage Inhibition	
0.0625 μM	33.5%	*p* = 0.003
0.125 μM	37.8%	*p* < 0.0001
0.25 μM	26.9%	*p* = 0.0015
0.5 μM	17.3%	*p* = 0.022
**Confocal Microscopy of Labelled α-Syn Seeds**		
R18D Concentration Examined	Labelled α-Syn Seed Uptake in Neurons—Percentage Inhibition	
0.125 μM	77.74%	*p* < 0.001
0.25 μM	76.0%	*p* < 0.001
0.5 μM	67.0%	*p* < 0.001

## Data Availability

The data presented in this study are available upon reasonable request from the corresponding author.

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
