# Peer review of "Novel Poly-Arginine Peptide R18D Reduces α-Synuclein Aggregation and Uptake of α-Synuclein Seeds in Cortical Neurons"

_biomedicines, 2025, doi:10.3390/biomedicines13010122_

Round 1

Reviewer 1 Report

Comments and Suggestions for Authors

This is a well-structured research article about the interesting question whether novel poly-arginine peptide r18d reduces a-synuclein aggregation and uptake of a-synuclein seeds in cortical neurons.

The introduction section is descriptive enough setting the background of the present paper, defining and analyzing PD, a-synucleinopathies and cationic arginine-rich peptides (CARPs).

Methodology section describes the methodology followed in the current study regarding study design, cell culture, cell assays and microscopic analysis.

Results are quite interesting and well depicted in images and graphs. However, I believe that some tables with the results’ data could further improve the understanding for the potential readers of the paper.

Discussion and conclusions are very well written including study’s limitations and potential targets for future studies.

Reviewer 2 Report

Comments and Suggestions for Authors

1. In the introduction, although it is stated in general terms, explicitly mention the drawbacks of existing treatments such as antibody and receptor antagonist therapies and then mention how R18D could make a significant difference in clinical treatment to overcome the drawbacks of existing treatments.

2. In the methods section, please specify how controls account for background signals in fluorescence assays like ThT and HTRF to ensure rigor.

3. Regarding microscopic imaging parameters, add more about resolution and signal sensitivity of parameters like z-stack step size and image processing techniques to alpha syn.

4. The article didn't mention if process of data collection was blinded. This may lead to observer bias and affect the accuracy of results. Mention this in the limitation section.

5. Mention why HTRF was specifically used to detect alpha syn aggregates rather than other methods. 

6. In section 2.2, why 2 hours for uptake, 48 hours for aggregation were used regarding R18D incubation.  Moreover, PD is a progressive disorder then why short term exposure rather than longer duration were used.

7. No control groups are mentioned that could differentiate effects from treatment with R18D peptide treatment from treatment with the peptide vehicle controls and/or potential intrinsic aggregation reduction by culture conditions and sonication.

8. Only LDH and MTS assays were performed to assess toxicity; other more specific assays, like oxidative stress markers or mitochondrial viability indicators, were not considered. (PMID: 39224576, PMID: 38464024)

9. Section 3.1, while electron microscopy provides qualitative data (presence or absent of aggregates). Providing quantitative measurements could improve the accuracy and understanding of the results and provide paths to statistically analyze the obtained data regarding aggregates (comparison with other studies, effect size ... etc). Mention this as a limitation.

10. In section 3.4, the study mentions detecting intracellular aggregates, however, it doesn't localize them (cytoplasmic vs nuclear).

11. It is unclear from the discussion how α-syn aggregation inhibition relates to functional improvements in neuronal health or clinical biomarkers of illness development (such CSF α-syn levels). suggest in your discussion that future research measure biomarkers in order to determine whether the effects of R18D correspond with disease biomarkers and can be used to predict clinical outcomes.

12. While R18D is represented to inhibit uptake and aggregation, the paper does not indicate through which pathways or mechanisms this takes place. For instance, does R18D block certain receptor-mediated pathways, such as heparan sulfate proteoglycans?
